# What Guides Peripheral Immune Cells into the Central Nervous System?

**DOI:** 10.3390/cells10082041

**Published:** 2021-08-10

**Authors:** Theresa Greiner, Markus Kipp

**Affiliations:** Institute of Anatomy, Rostock University Medical Center, 18057 Rostock, Germany; theresa.greiner@uni-rostock.de

**Keywords:** cuprizone, experimental autoimmune encephalomyelitis, multiple sclerosis, inflammation

## Abstract

Multiple sclerosis (MS), an immune-mediated demyelinating disease of the central nervous system (CNS), initially presents with a relapsing-remitting disease course. During this early stage of the disease, leukocytes cross the blood–brain barrier to drive the formation of focal demyelinating plaques. Disease-modifying agents that modulate or suppress the peripheral immune system provide a therapeutic benefit during relapsing-remitting MS (RRMS). The majority of individuals with RRMS ultimately enter a secondary progressive disease stage with a progressive accumulation of neurologic deficits. The cellular and molecular basis for this transition is unclear and the role of inflammation during the secondary progressive disease stage is a subject of intense and controversial debate. In this review article, we discuss the following main hypothesis: during both disease stages, peripheral immune cells are triggered by CNS-intrinsic stimuli to invade the brain parenchyma. Furthermore, we outline the different neuroanatomical routes by which peripheral immune cells might migrate from the periphery into the CNS.

## 1. Introduction

There are, basically, two clinical courses of multiple sclerosis (MS): a relapsing and a chronically progressive course. Relapsing-remitting MS (RRMS) is characterized by recurrent episodes of clearly delineated clinical impairment (i.e., relapsing) from which the patients can recover completely (i.e., remission). In many cases, a relapsing-remitting disease course turns into a secondary progressive (SPMS) course after approximately 10–15 years. In this secondary phase of the disease, individual relapses occur less frequently but there is a slow, more or less continuous progression of the clinical impairment. In several patients, the beginning of the disease is not characterized by relapses but an accumulation of clinical disabilities occurs. This is referred to as primarily progressive MS (PPMS). According to today’s opinion, the relapses are based on a focal inflammatory demyelination of the white matter. Peripheral immune cells, especially T- and B-lymphocytes as well as monocytes, which migrate en masse into the central nervous system (CNS), seem to be decisively involved in inflammatory lesion development. During the progressive course, on the other hand, diffuse tissue damage to the white and gray matter—in particular, a destruction of axons, synapses or even entire nerve cells—is mainly responsible for the progressive clinical impairment [1,2].

According to our current understanding, peripheral immune cells are involved in both phases of the disease, relapsing and progressive MS. Extensive pathological studies have shown that in progressive MS the extent of axonal damage correlates positively with the density of intracerebral lymphocytes [3]. In addition, histological correlates of lymphoid tissue with a high B-cell density are regularly found in the meninges of deceased progressive MS patients [4,5,6]. From these and other findings it can be deduced that, during progressive MS, peripheral immune cells are still functionally involved in progressive tissue destruction. Although immunosuppressive drugs can reduce the relapse frequency during the secondary progressive disease stage [7], it is entirely unclear why classical immunosuppressive drugs such as natalizumab do not ameliorate the disease progression in the progressive stage of the disease [8]. Possible explanations are (i) that intracerebral inflammatory mechanisms in progressive MS become chronic and establish themselves within the CNS, (ii) that not peripheral immune cells but CNS-resident immune cells (i.e., microglia and/or astrocytes) significantly influence disease progression in this phase, (iii) that peripheral immune cell recruitment still occurs during the progressive disease stage but the phenotype of the recruited immune cells changes during the course of the disease and can, thus, no longer be influenced by classic immunomodulatory drugs or (iv) that different mechanisms or neuroanatomical pathways are involved in immune cell recruitment in relapsing versus progressive MS. In this context, it is important to note that during the progressive disease stage natalizumab does not completely prevent the immune cells from entering the CNS but favors the accumulation of plasma cells [9]. Again, the pathogenic and clinical significance of this cellular transformation is not known.

## 2. Inflammation in Multiple Sclerosis

In MS, acute inflammatory lesions are, on the histopathological level, characterized by a focal loss of the myelin sheaths (i.e., demyelination), oligodendrocyte loss and, to a variable extent, axonal degeneration. Astrocytes and microglia cells are activated and peripheral immune cells, among lymphocytes and monocytes, invade the affected and surrounding tissue areas. The sequence of cellular and molecular events leading to the development of oligodendrocyte damage and demyelination are not fully understood. Factors involved that potentially lead to oligodendrocyte destruction are oxidative stress, mitochondrial dysfunction, nitric oxide accumulation, protein misfolding or inflammatory cytokine exposure [10,11]. Of note, oligodendrocytes and an intact myelin sheath are important for neuronal health. Firstly, oligodendrocytes provide nutritional support to the neurons [12] and secondly, the proper functioning of the fast axonal transport critically depends on viable oligodendrocytes [13]. Consequently, mice deficient in mature myelin proteins display severe neurodegeneration [14].

The basis for the initial diagnosis and monitoring of the course of MS continues to be the detection of lesions in the brain using native T1- and T2-weighted images or fluid attenuated inversion recovery (FLAIR) sequences as well as contrast-enhanced T1-weighted sequences. During imaging, the administration of contrast agents can differentiate acute, mostly contrast-enhancing, lesions from chronic lesions. While inflammation appears to play a role during all stages of the disease [3], the extent of inflammation is lower in progressive compared to RRMS. Between SPMS and PPMS patients, differences with respect to the inflammatory activity also exist. It has been shown, for example, that enhancing lesions are less frequent in PPMS compared to SPMS [15]. In this study, twelve SPMS and twelve PPMS patients were followed by gadolinium diethylenetriaminepentaacetic acid (Gd-DTPA)-enhanced MRI over a six-month period. Patients in the secondary progressive group had a total of 109 new lesions over this time (equals ~18.2 lesions per patient per year) and most of these were enhanced. In contrast, only 20 new lesions were seen in the primary progressive group (~3 lesions per patient per year) and only one of these was enhanced. Despite the greater number of enhancing lesions, there was no difference in the degree of clinical deterioration between the two groups over the six-month study period, underpinning the view that clinical worsening is mainly driven by neuro-axonal degeneration rather than inflammation. In agreement with these results, others have shown higher T1 and T2 lesion loads in SPMS versus PPMS [16,17]. Higher inflammatory activities in SPMS versus PPMS have also been observed in post-mortem studies. Revesz and colleagues showed, by histologically analyzing five hundred and seventy-eight lesions, that there is significantly more inflammation in SPMS (as judged by the frequency of perivascular cuffing and cellularity of the parenchyma) than in PPMS [18]. Interestingly, T-cells isolated from PPMS patients with a relatively high T2 lesion load demonstrated a greater migratory potency and cytokine production compared to T-cells isolated from PPMS patients with a relatively low T2 lesion load [19]. This suggests that the higher inflammatory activity might also be due to a higher migration potential of peripheral immune cells, at least in PPMS patients. To conclude, although distinct differences exist between different MS courses, inflammation is an important element during all disease stages.

Of note, the neuroanatomical compartments where lesions develop are different during the different disease stages. As demonstrated by Haider and colleagues, cortical demyelination is most pronounced in progressive MS whereas during RRMS, periventricular lesions are the most frequent lesion type [20]. Another study showed that in the spinal cord, the central cord area was more often affected by lesions in PPMS than RRMS patients [21]. Meningeal inflammation in the form of ectopic lymphoid-like structures is (probably) found more frequently in SPMS compared to RRMS patients [4,5]. The site of the peripheral immune cell recruitment (see above) might, thus, change during the disease progression. This might be paralleled by functional changes of the immune cells. In support of this, patients with exclusively cerebral or preferential spinal lesion manifestation were associated with increased proportions of circulating granulocyte-macrophage colony-stimulating factor-producing TH_1_ cells or interleukin (IL)-17-producing TH_17_ cells, respectively. In contrast, the proportions of the peripheral IL-17/IL-22-producing lymphoid tissue inducer, the innate counterpart of TH_17_ cells, were enhanced in RRMS patients with exclusively a cerebral lesion topography [22].

## 3. Along Which Neuroanatomical Pathways Do Peripheral Immune Cells Travel inside the CNS?

It is currently unclear what guides peripheral immune cells from the vasculature into the brain parenchyma. The two main principal neuroanatomical routes that peripheral immune cells use to invade the CNS are (i) immune cell diapedesis across the blood–brain barrier and (ii) immune cell trafficking across the choroid plexus. Lymphocyte migration across the blood–brain barrier, which occurs at the level of the post-capillary venules, represents a multi-step process starting with selectin-dependent lymphocyte vascular rolling that initiates the contact between the lymphocyte and the endothelium (for detailed reviews, see [23,24]). Rolling serves as a break for circulating leukocytes and is mediated by E-, P- and L-selectins, expressed by both “partners”, the leukocytes and the endothelial cells. Rolling reduces the speed of the immune cells allowing for their subsequent recognition of endothelial chemokines with their G-protein-coupled receptors (GPCRs). This leads to the activation of an intracellular signaling cascade, which in turn activates the adhesion molecules of the integrin family leading to firm leukocyte adhesion/arrest on the luminal surface of the endothelial cells. This allows the immune cells to find the endothelial junctions, enabling their diapedesis across the endothelial barrier towards the subendothelium. A number of molecular players are known to mediate this multi-step process. For example, integrin α4β1 (very late antigen-4; VLA-4), expressed by immune cells, can interact with vascular cell adhesion molecule 1 (VCAM-1), expressed by endothelial cells [25,26,27], to mediate leucocyte recruitment into the CNS. The established MS drug natalizumab, a monoclonal antibody that acts as an α4 integrin antagonist, prevents leukocyte trafficking into the CNS by interfering with this pathway. As will be pointed out later, brain-intrinsic degenerative and/or inflammatory processes can trigger the activation of the signaling cascades involved during peripheral immune cell recruitment over the blood–brain barrier. For example, glial cells are an important source of the pro-inflammatory chemokines CCL2/MCP-1, RANTES and CXCL10/IP-10, which are required for TH_1_ and TH_17_ lymphocyte and monocyte recruitment into the CNS [28,29].

The second neuroanatomical pathway peripheral immune cells might use to reach the brain parenchyma is from the stroma of the choroid plexus, through the choroid plexus epithelial cell layer into the ventricles and from there through the ependymal cell layer into the periventricular brain regions (see Figure 1A,B). The choroid plexus, the main producer of cerebrospinal fluid (CSF), is located in each of the brain ventricles and consists of a highly vascularized stroma surrounded by a tight continuous layer of epithelial cells resembling specialized ependymal cells. These specialized ependymal cells are rich in microvilli and a few kinocilia, which guarantee a high level of liquor secretion and directed liquor transport. In contrast to the endothelial cells of the brain vessels, the vasculature of the choroid plexus is characteristically fenestrated and not continuous, resulting in a leaky interphase between the blood compartment and the choroid plexus stroma. However, the epithelium of the choroid plexus is equipped with a dense network of tight junctions, restricting the entry of molecules and cells from the stroma into the inner CSF spaces. The tight junction of the choroid plexus epithelial cell layer is, thus, a key component of what is commonly called the blood–CSF barrier (BCSFB). The BCSFB is only permeable to water and the gases dissolved in it. All other (macromolecular) components of the CSF are secreted through highly selective transport mechanisms. Due to these properties of the BCSFB, the liquor is poor in proteins and cells. The glucose level is 30–50% below that of the serum blood sugar level. The cellular and biochemical composition of the CSF often changes as a result of the pathological processes within the CNS. Indeed, the detection of inflammatory changes within the CSF plays a central role in the diagnosis of MS. Before the introduction of the various methods for the detection of oligoclonal immunoglobulin bands, the detection of lymphocytic pleocytosis, which is commonly found in MS patients, was of great importance for an MS diagnosis [30].

In an autoimmune MS mouse model experimental autoimmune encephalomyelitis (EAE), the choroid plexus has been shown to be an important early entry point for immune cells into the CNS [31]. Furthermore, post-mortem studies showed a pro-inflammatory milieu in the choroid plexus of MS patients including increased numbers of macrophages, dendritic cells, granulocytes and CD8^+^ T-cells [32,33]. In addition, another post-mortem study showed a loss of the key tight junction protein claudin-3 at the choroid plexus in MS patients compared to control tissues [34]. Together, these studies suggest that the choroid plexus might be an important mediator of inflammatory responses as well as an entry point of peripheral immune cells into the CNS of MS patients.

Less well-investigated is a third potential migration pathway of peripheral immune cells into the brain parenchyma, namely, via leptomeningeal vessels into the subarachnoid space and from there through the glia limitans superficialis directly into the brain parenchyma (see Figure 1C). Bartholomäus and colleagues experimentally addressed this migration route at the level of the spinal cord during EAE in rats and were able to demonstrate that T-cells specific for the mature myelin protein myelin basic protein arrived at the CNS at the level of the leptomeninges of the spinal cord before they entered the parenchyma with the onset of the clinical disease [35]. The same group later demonstrated a rapid turnover of T-cells in this CNS compartment, showing that these leptomeningeal T cells display an activated phenotype and that VLA-4 and/or LFA-1 are functionally involved in leptomeningeal immune cell recruitment [36].

A fourth potential migratory pathway of peripheral immune cells into the brain parenchyma might be via leptomeningeal vessels into the subarachnoid space and from there along the penetrating brain vessels, driven by the glymphatic flow, into the depths of the brain (see Figure 1D). Such a migration route could imitate the appearance of a perivascular cuff in an isolated paraffin section and, thus, might have been underestimated in the past. The results of a previous study suggest that at the level of the spinal cord such a migration route is not common. After an injection of encephalitogenic T-cells into the subarachnoid space of the cisterna magna or the lumbar spinal cord, the majority of cells accumulated at the levels of the cell injections without traveling along the perivascular spaces [36]. However, the results of our lab suggest that the perivascular spaces of the penetrating arteries might indeed be an entry point of peripheral immune cells into the brain. In Cup/EAE mice (for a description of the model, see the next section) a gradual increase of T-cell densities could be observed in the perivascular spaces of the large penetrating arteries in mice. As demonstrated in Figure 2, at more superficial parts of the penetrating artery, T-cells are located predominantly within the two basal laminae (i.e., endothelial and astrocytic basal lamina), which corresponds with the Virchow–Robin space. In deeper parts of the penetrating vessel, T-cells can be found in equal numbers inside the Virchow–Robin space and around the vessel wall. Our data, thus, support the view that the leptomeninges represent an important checkpoint for T-cell infiltration during autoimmune inflammation within the CNS.

## 4. What Triggers Peripheral Immune Cell Recruitment?

Extensive pathological and imaging studies suggest that MS lesions tend to develop in previously damaged or pre-damaged areas of the CNS [37,38,39]. Barnett and Prineas showed that classical active demyelinating lesions can start with oligodendrocyte degeneration paralleled by focal microglia activation. T-cells and activated peripheral macrophages later populate the developing lesion [37]. Similar findings have been reported by others although different terminologies have been used to describe such pathological alterations [40,41,42]. Various radiological approaches provide evidence that subtle, non-inflammatory alterations in the normally appearing white matter of MS patients precede the development of classical inflammatory foci. For example, Filippi and colleagues performed magnetization transfer scans in 10 MS patients every 4 weeks for 3 months with Gd-DTPA injections to visualize the blood–brain barrier integrity. Weeks before a Gd-DTPA enhancement appearance, the mean magnetization transfer values in the normally appearing white matter, measured from areas corresponding with future enhancing lesions, were significantly lower than the mean magnetization transfer values in the normally appearing white matter outside enhancing areas [43]. EAE in vivo experiments, conducted by Maggi and colleagues in marmosets by combining serial in vivo MRI and post-mortem analyses, showed that early EAE lesions show focal microglia and astrocyte activation in the absence of demyelination and parenchymal lymphocytes [44]. These and other findings [45,46] clearly demonstrate that new focal lesions associated with a frank blood–brain barrier leakage are preceded by subtle, progressive alterations in the tissue integrity beyond the resolution of conventional MRI.

In recent years, our lab has contributed toward the testing of the hypothesis that a brain-intrinsic degenerative process can trigger peripheral immune cell recruitment into the CNS parenchyma. In our studies, we took advantage from two different MS animal models, the toxin-induced demyelination model cuprizone and the autoimmune EAE model. In the cuprizone model, which is characterized by oligodendrocyte apoptosis followed by demyelination, different white and grey matter forebrain regions are affected among the corpus callosum [47], the cortex [48,49] and the hippocampus [50] whereas the spinal cord is spared [51]. In the EAE model, inflammatory demyelination is prominent in the spinal cord and cerebellum whereas the forebrain is not or only moderately involved. To follow the hypothesis that brain-intrinsic degenerative processes are a sufficient and potent trigger for peripheral immune cell recruitment, demyelination in murine forebrains was induced by a three-week cuprizone intoxication period, which was followed by a period of two weeks on normal chow. At the end of week five, autoreactive T-cell development was experimentally induced by active immunization with the myelin oligodendrocyte glycoprotein 35–55 peptide (MOG_35–55_) dissolved in Complete Freund’s Adjuvant [52]. While the peripheral immune cell recruitment was minimal in the forebrains of MOG_35–55_-immunized mice (i.e., active EAE induction), the inflammatory demyelination of the forebrain was widespread in Cuprizone + EAE mice (i.e., Cup/EAE). These initial experiments demonstrated that cuprizone-induced demyelination can act as a potent trigger for peripheral immune cell recruitment [53,54]. A limitation of these first studies was, however, that active MOG_35–55_ immunization was induced at a time point when demyelination was already fully established. This series of cellular events does not reflect the proposed mechanisms operant during the initial MS lesion formation, i.e., oligodendrocyte degeneration and microglial activation associated with few lymphocytes and phagocytes in the regions of relative myelin preservation [37,38,55,56]. In a follow-up study, we modified the protocol and mice were intoxicated with cuprizone for one week and then immunized with the MOG_35–55_ peptide [57]. After one week of cuprizone intoxication, the corpus callosum of the experimental animals showed several histopathological characteristics of early MS lesions among the focal microglia activation and oligodendrocyte degeneration but an absence of lymphocytes and overt demyelination. We showed that this relatively mild pathological process indeed triggers peripheral immune cell recruitment. Similar findings have been reported by others. For example, Baxi and colleagues showed that transferred encephalitogenic TH_1_ and TH_17_ CD4^+^ T-cells infiltrate the CNS of cuprizone-intoxicated mice [58]. These findings demonstrate that autoreactive immune cells, if present in the peripheral circulation, are recruited to the sites of the pre-injured brain parenchyma. We hypothesize that this occurs via a three-step process. First, cuprizone activates the cells of the blood–brain barrier (i.e., astrocytes, pericytes and/or endothelial cells), compromising its integrity and activating chemokine-dependent leukocyte recruitment pathways (see above). Second, encephalitogenic immune cells invade the forebrain parenchyma and are locally reactivated. Third, a secondary wave of immune cell recruitment leads to inflammatory demyelinating lesions.

## 5. Does a Degenerative Process in the CNS Trigger Autoimmunity?

Several excellent review articles have discussed the cellular signature of neurodegeneration in MS [59,60,61,62] and basic evidence exists that peripheral immune cells can either promote [63,64,65] or ameliorate [66,67,68] neurodegeneration. While it is clear that peripheral encephalitogenic immune cells can be triggered by central stimuli to invade the brain, it is highly debated whether the degeneration of CNS elements per se is sufficient to trigger autoimmune responses. In an elegant experiment, Caprariello and colleagues intoxicated mice with cuprizone for two weeks, a period sufficiently brief to perturb the myelin ultrastructure without causing overt demyelination. This cuprizone “induction phase” was then followed by an artificial immune stimulus with a complete Freund’s adjuvant and pertussis toxin. Unlike during the induction of active EAE, exogenous myelin peptides (for example, MOG_35–55_) were excluded from the adjuvant in their experiments so that the immunogenicity of endogenous cuprizone-altered myelin could be directly tested. Cuprizone intoxication together with the immune stimulus resulted in a severe inflammation as demonstrated by high densities of CD3^+^ lymphocytes, IBA1^+^ microglia/macrophages and GFAP^+^ astrocytes [69]. The fact that degenerative events inside the CNS can trigger the recruitment of peripheral immune cells into the CNS has been shown as well in other experimental paradigms such as in a model of a peripheral facial nerve injury [70], a cortical cryoinjury or an eyeball enucleation [71]. In the archetypical neurodegenerative disorder amyotrophic lateral sclerosis (ALS), the recruitment of T-cells is well known [72]. Lymphocytes were found in the majority of ALS patient spinal cords and along the vessels in the precentral gyrus extending into the areas of neuronal injury [73,74]. T-helper cells were found in proximity to degenerating corticospinal tracts while T-suppressor/cytotoxic cells were demonstrated in the ventral horns [75].

In a recent study, we were able to show that, during the course of acute cuprizone-induced demyelination, the recruitment of lymphocytes can be observed [76]. Interestingly, in contrast to most EAE models where inflammatory infiltrates are dominated by CD4^+^ T-cells [77], CD8^+^ cytotoxic T-cells outnumbered CD4^+^ T-helper cells. Of note, the densities of the recruited lymphocytes were found to be comparable in the cuprizone model and the lesions obtained from progressive MS patients (see Figure 3). It is currently not clear whether these observed T-cells are indeed activated. Previous studies have shown that resting lymphocytes do not breach the blood–brain barrier. In contrast, activated T-lymphocytes can cross the blood–brain barrier regardless of their antigen specificity but only T-cells that recognize the CNS antigen persist and can recruit other inflammatory cells [78]. The expression of the proliferation markers KI-67 and PCNA as well as the polarized expression of the cytolytic granule protein Granzyme B in cuprizone-intoxicated mice indicates that the recruited T-cells are somewhat activated. However, peripheral monocyte recruitment is sparse in the cuprizone model [54], indicating that the recruited T-cells in the cuprizone model do not display the full activation program to trigger the massive second wave recruitment of other immune cells. Of note, only T-cells and no B-cells were found in cuprizone tissues, which is in line with findings in ALS patients. Further studies have now to show whether the CNS-triggered peripheral immune cell recruitment favors T-cells over B-cells to invade the CNS and, if so, which signaling pathways are operant.

## 6. Conclusions

Local resident inflammatory cells, id est astrocytes and microglia, are equipped with the molecular machinery to express and secrete chemokines and other pro-inflammatory cytokines [79,80,81,82]. Such cytokines can participate in the activation of endothelial cells, paving the way for peripheral immune cell recruitment into the brain parenchyma. It is our strong belief that such cell–cell communication pathways play a pivotal role during peripheral immune cell recruitment in MS as well as in other neurodegenerative disorders. A better understanding of which molecular pathways are involved during this process will pave the way for the development of novel therapeutic strategies in the future.

## Figures and Tables

**Figure 1 cells-10-02041-f001:**
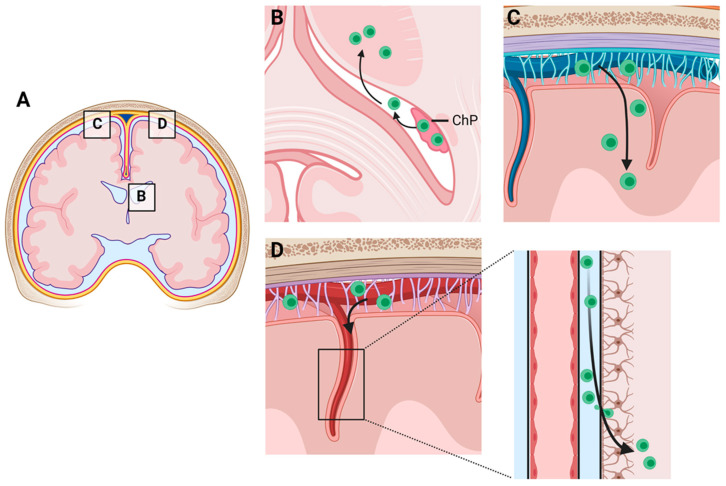
Schematic illustration of alternative routes for peripheral immune cell recruitment into the central nervous system. (**A**) Overview. (**B**) Immune cell recruitment across the blood–CSF barrier, also called the blood–liquor barrier. This barrier restricts the passage of substances from the blood into the liquor. (**C**) Immune cell recruitment via leptomeningeal vessels into the subarachnoid space and from there through the glia limitans superficialis directly into the brain parenchyma. (**D**) Alternatively, immune cells within the subarachnoid space can migrate along the penetrating brain vessels, driven by the glymphatic flow, into the depths of the brain. See Figure 2 for experimental proof of this migration route. (Created with BioRender.com, access date on 5 August 2021).

**Figure 2 cells-10-02041-f002:**
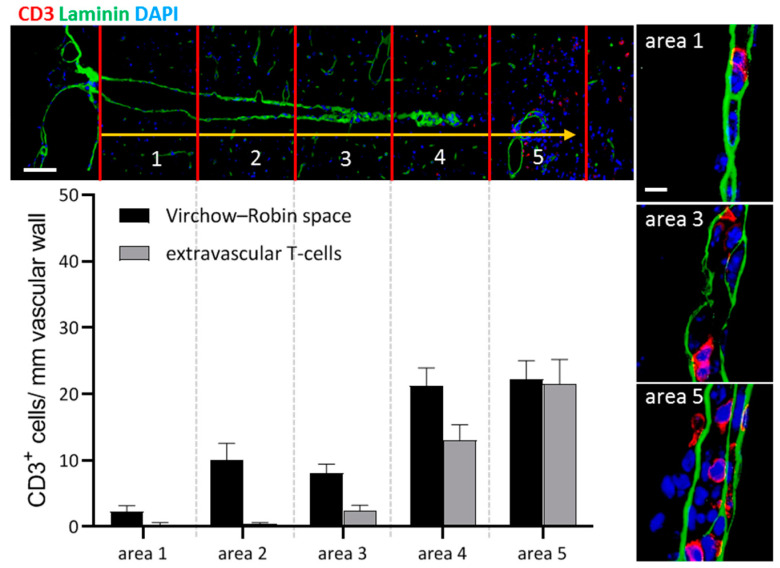
CD3^+^ lymphocytes (red) along a penetrating blood vessel in Cup/EAE mice. The brain surface is shown on the left. Area 1 represents the most superficial part of the neocortex whereas Area 5 is adjacent to the white matter tract corpus callosum. The perivascular Virchow–Robin space is demarcated by the two basal laminae (laminin^+^; green). Note the increasing cell densities towards the depth of the brain. Scale bars: upper left: 100 µm, right: 10 µm.

**Figure 3 cells-10-02041-f003:**
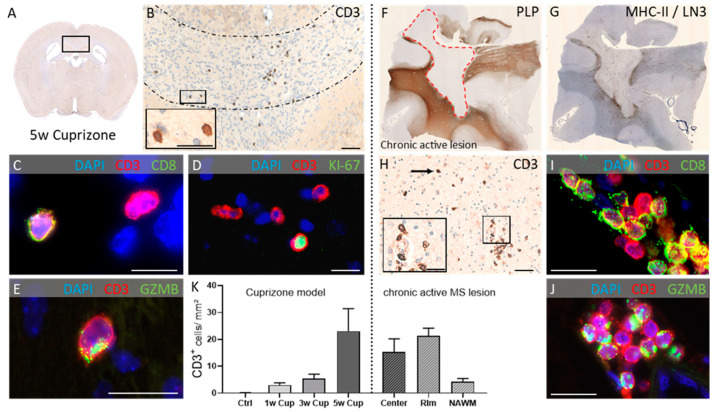
Peripheral immune cell recruitment in the cuprizone model (left) and progressive MS lesions (right). (**A**) Overview. (**B**) Representative image demonstrating CD3^+^ lymphocytes in the corpus callosum of cuprizone-intoxicated mice (five weeks). Around 60 % of the recruited lymphocytes are CD8^+^ cytotoxic lymphocytes. (**C**) ~20–30 % are proliferating (Ki67^+^; (**D**)) and express cytotoxic granules (GZMB^+^; (**E**)). (**F**) Overview of a representative chronic active multiple sclerosis (MS) lesion. The red dotted line demarcates the lesion in an anti-myelin proteolipid protein (PLP)-stained section. (**G**) A consecutive anti-major histocompatibility complex class II (MHC-II)/LN3-stained section shows the lesion rim. (**H**) High-power view of the same lesion demonstrates the distribution of CD3^+^ lymphocytes. Lymphocytes display a CD8+ phenotype (**I**) and express cytotoxic granules (GZMB^+^; (**J**)). (**K**) Comparative densities of CD3^+^ lymphocytes during the course of cuprizone-induced demyelination and in the center, rim and normally appearing white matter (NAWM) areas of chronic active MS lesions. PLP: Proteolipid-Protein 1; DAPI: 4′,6-Diamidino-2-phenylindol; GZMB: Granzyme B. Scale bars: (**B**) = 50 µm, insert (**B**) = 20 µm, (**C**) = 10 µm, (**D**) = 15 µm, (**E**) = 10 µm, (**H**) = 50 µm, insert (**H**) = 25 µm, (**I**) = 10 µm, (**J**) = 10 µm. Figure modified, courtesy of John Wiley & Sons, license number: 5084321153999.

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
