# Peer review of "What Guides Peripheral Immune Cells into the Central Nervous System?"

_cells, 2021, doi:10.3390/cells10082041_

Round 1

Reviewer 1 Report

This review is clear and exhaustive, with pertinent figures. The authors described immune cell infiltrates into the SNC at early and later stages of MS, comparing relapsing-remitting MS to primary and secondary progressive MS. This review brings the latest findings concerning pathological mechanisms of inflammation.

Some points must be addressed:

The authors, claims that: “However, it is entirely unclear why classical immunosuppressive drugs, such as Natalizumab, lose their effectiveness in the progressive stage of the disease”, page 2. There is still a debate concerning the efficiency of these drugs in progressive MS. Roos I et al (Neurology 2021) showed in a retrospective cohort that high immunosuppressive drugs suh as Natalizumab reduced relapse numbers in progressive MS. Moreover, Hausler D et al (Brain Pathol 2021) described modifications of immune cell in SNC from Natalizumab treated patients. This point should be more detailed.

Page 2, line 64: “microglial cells aret” instead of “are”

Page 2, line 67, a missing word “are”: “the sequence… not fully understood”.

The section: author contribution was not filled.

Author Response

Reviewer 1

Q1: The authors, claims that: “However, it is entirely unclear why classical immunosuppressive drugs, such as Natalizumab, lose their effectiveness in the progressive stage of the disease”, There is still a debate concerning the efficiency of these drugs in progressive MS. Roos I et al (Neurology 2021) showed in a retrospective cohort that high immunosuppressive drugs such as Natalizumab reduced relapse numbers in progressive MS. Moreover, Hausler D et al (Brain Pathol 2021) described modifications of immune cell in SNC from Natalizumab treated patients. This point should be more detailed.

A1: Thank you for this instructive comment. We have revised the paragraph accordingly and mention now both studies in the revised version of the manuscript.

Q2: Page 2, line 64: “microglial cells aret” instead of “are”; Page 2, line 67, a missing word “are”: “the sequence… not fully understood”.

A2: Thank you for the notes. We have revised the manuscript accordingly.

Q3: The section: author contribution was not filled.

A3: Thank you for this comment. We have modified this part.

Reviewer 2 Report

The Authors address and discuss an important aspect of multiple sclerosis (MS) pathogenesis: the peripheral immune cell invasion of the central nervous system. The Authors are experts in the field with several high impact publications on MS and immune cells. The manuscript is well-written, has a logical structure with a good balance of outlining basic concept and discussing hypothesis and novel developments in the field.

I have the following specific comments and suggestions:

  • Discuss in more details neurodegeneration in multiple sclerosis. Subheading 5 touches upon this, and mentions ALS as an example that in a neurodegenerative disease T-lymphocyte recruitment occurs. However, it is noteworthy that in MS neurodegenerative processes are in place, and these are not unrelated to the immune-mediated processes.
  • Title: Add _?_ to the end
  • Line 9: delete _ ; _
  • Abstract: line 10: MS is an immune mediated demyelinating disease (instead of _inflammatory_)
  • Line 15: correct type in _progressive_
  • Line 18: discuss is suggested instead of _raise_
  • Line 62: demyelination is suggested onstead of _inflammatory_
  • Line 159: Format: Figure 1A,B (also elsewhere in the manuscript)
  • Lines 343-349: Author contributions missing
  • line 358: specify what _kind support_ means
  • References: Apply Cells style. use of capital letters in journals’ names
  • Line 559: delete _1_

Author Response

Q1: Discuss in more details neurodegeneration in multiple sclerosis. Subheading 5 touches upon this, and mentions ALS as an example that in a neurodegenerative disease T-lymphocyte recruitment occurs. However, it is noteworthy that in MS neurodegenerative processes are in place, and these are not unrelated to the immune-mediated processes.

A1: Thank you for this fruitful comment, we have addressed this point in the revised version of the manuscript at the beginning of chapter 5.

Q2: Title: Add _?_ to the end; Line 9: delete _ ; _; Abstract: line 10: MS is an immune mediated demyelinating disease (instead of _inflammatory_); Line 15: correct type in _progressive_; Line 18: discuss is suggested instead of _raise_; Line 62: demyelination is suggested onstead of _inflammatory_; Line 159: Format: Figure 1A,B (also elsewhere in the manuscript); Lines 343-349: Author contributions missing; Line 358: specify what _kind support_ means; References: Apply Cells style. use of capital letters in journals’ names; Line 559: delete _1_

A2: Thank you for these notes. We have adopted the manuscript accordingly. Beyond, we have downloaded the latest MDPI Journal style and hope to meet now the criteria of the fine journal.